# Long-Term Evolution of the Caspian Sea Thermohaline Properties Reconstructed in an Eddy-Resolving OGCM

Gleb S. Dyakonov[1,2], Rashit A. Ibrayev[1,2,3]

[1]Northern Water Problems Institute, Russian Academy of Sciences, Petrozavodsk, Russia
[2]Shirshov Institute of Oceanology, Russian Academy of Sciences, Moscow, Russia
[3]Marchuk Institute of Numerical Mathematics, Russian Academy of Sciences, Moscow, Russia

*Correspondence to:* Gleb S. Dyakonov (gleb.gosm@gmail.com)

**Abstract.** Decadal variability of the Caspian Sea thermohaline properties is investigated using a high-resolution ocean general circulation model including sea ice thermodynamics and air-sea interaction forced by prescribed realistic atmospheric conditions and riverine runoff. The model describes synoptic, seasonal and climatic variations of sea thermohaline structure, water balance and sea level. A reconstruction experiment was conducted for the period of 1961-2001, covering a major regime shift in the global climate during 1976-1978, which allowed investigating the Caspian Sea response to such significant episodes of climate variability. The model reproduced sea level evolution reasonably well despite that many factors (such as possible seabed changes and yet insufficiently explored underground water infiltration) were not taken into account in the numerical reconstruction. This supports the hypothesis relating rapid Caspian Sea level rise in 1978-1995 with the global climate change, which caused variation of local atmospheric conditions and riverine discharge reflected in the used external forcing data, as is shown in the paper. Other effects of the climatic shift are investigated including a decrease of salinity in the active layer, strengthening of its stratification and corresponding diminishing of convection. It is also demonstrated that water exchange between the three Caspian basins (northern, middle and southern) plays a crucial role in the formation of their thermohaline regime. The reconstructed long-term trends in sea water salinity (general downtrend after 1978), temperature (overall increase) and density (general downtrend) are studied, including an assessment of the influence of main surface circulation patterns and model error accumulation.

## 1 Introduction

The Caspian Sea is the largest enclosed water body on earth with a surface area of more than 370 000 km² and a catchment area almost 10 times greater. Yet it is highly sensitive to variations in the global and regional climate systems as well as economic activities that include major schemes of river regulation. This is vividly reflected in the evolution of the Caspian Sea level, which is subject to large fluctuations on seasonal and decadal timescales. The water balance of the isolated sea varies significantly due to the seasonal character of the riverine discharge, accounting for sea level oscillations with an amplitude of 20–40 cm. Long-term fluctuations of the level are even larger: in the second half of the 20th century they amounted to 2.5 m.

Prediction of the long term impacts of climate change and man-made activities on the Caspian represents a great scientific challenge important for fisheries, coastal development and other industries of the region. Ocean general circulation models (OGCM) have greatly advanced our understanding of the Caspian Sea circulation patterns, particularly its seasonal variability (Arpe et al., 1999; Ibrayev, 2008; Kara et al., 2010; Ibrayev et al., 2010, Gündüz and Özsoy, 2014, Diansky et al., 2016). The increasing production of global atmospheric reanalysis datasets and their availability over several decades have made possible retrospective studies of the long-term evolution of the marine environment, based on numerical reconstruction of its response to external forcing, as will be done in the present paper. This approach was applied in our previous work (Dyakonov and Ibrayev, 2018) with emphasis on the long-term variability of the Caspian Sea water balance and its sensitivity to external factors. Now we use the same model to study the evolution of thermohaline properties (temperature, salinity and density) of the Caspian Sea in 1961-2001. The period is particularly interesting, as it covers one of the most notable events of global climate change – the climate shift of 1976-1978, also referred to as the Great Pacific Climate Shift, widely discussed in literature (Miller et al., 1994; Wooster and Zhang, 2004; Powell and Xu, 2011). The shift was associated with a change in major climatic indicators such as the North Atlantic Oscillation with significantly increased cyclonic activity and air humidity in Europe consequently leading to a sharp rise of the Caspian Sea level and changes in its stratification. The weakened ventilation of the deep sea, in turn, has led to degradation of the ecological situation in the sea (Tuzhilkin et al., 2011).

In the present paper we analyze long-term evolution of the Caspian Sea water parameters obtained in a numerical reconstruction experiment, which is described in section 2. In order to better understand model results, the evolution of the prescribed atmospheric and riverine forcing is briefly considered in section 3. In section 4 we discuss main patterns of surface circulation, which will help explaining further results. Then we proceed with model validation based on comparison of the obtained evolution of several in-situ parameters with observations (section 5). Finally, in section 6 we analyze long-term variability of thermohaline properties of the sea and its response to climatic variations. The Caspian Sea comprises three basins, partly separated by peninsulas extending into the sea interior: northern, middle and southern (respectively referred to as NorthCS, MidCS and SouthCS). Due to great differences between the basins in terms of bottom relief, non-uniform distribution of river run-off and large sea extent in the latitude direction, thermohaline circulation of each basin is distinctively different from others. Therefore, the analyzed properties of water masses have been averaged over a certain horizon for every Caspian basin separately. Averaging in the horizontal plane simplifies the analysis but conceals many subbasin-scale features of the fields, which must be kept in mind. In the following figures vertical dashed lines mark instances where climatic shifts occur.

## 2 Experiment setup

### 2.1 Model description

In Ibrayev (2001), Ibrayev et al. (2001; 2010) a three-dimensional primitive equation numerical model MESH (Model for Enclosed Sea Hydrodynamics) was presented, which was developed to study the Caspian Sea seasonal variability. The model successfully coped with this task and was used as a basis in the present research. However, an investigation of the Caspian Sea circulation on decadal timescale imposes additional requirements on the model, therefore it has been considerably redesigned. Geopotential vertical coordinate (z-coordinate), which had been used in MESH, was replaced by a hybrid system with a terrain-following sigma coordinate in the upper 30 m of the sea covering shallow regions, and a z-coordinate below 30 m depth. The long-term fluctuations of the Caspian Sea surface height (CSSH) are greater than the seasonal by an order of magnitude and could cause numerical instabilities and errors in a z-coordinate model. The use of a sigma coordinate ensures model stability during CSSH lows, allows much better resolution of surface boundary layer structure and diurnal air-sea interaction cycle during CSSH highs. A sigma coordinate grid also provides an accurate representation of the northern Caspian shelf bathymetry with increasingly flat slope (see Fig. 1). This is necessary to reconstruct the evolution of the sea surface area, essential for the evolution of air-sea fluxes in an environment of coastal flatlands, subject to large CSSH fluctuations. Additionally, the model has been equipped with a flooding/drying algorithm, enabling it to describe shoreline variations related to mean sea level change and wind surges. A more detailed description of the model used in this work is presented in Ibrayev and Dyakonov (2016); Dyakonov and Ibrayev (2016; 2018). Here we will note only the main points.

We use the Caspian Sea bathymetry based on the ETOPO1 dataset (Amante and Eakins, 2009), while particular attention has been paid to correctly interpolate the data onto the model grid and to preserve fine details such as islands and the shoreline. The Kara-Bogaz-Gol Bay was erased from the relief, as its connection to the sea is unilateral, and the corresponding boundary condition was set, to account for the outflow of sea water into the bay. The resulting bathymetry is presented in Fig. 1. The model has a resolution ~4.3 km in the horizontal plane, which is relatively high, as the Rossby baroclinic deformation radius is 17–22 km in deep-water areas of the Caspian (Arkhipkin et al., 1992). Eddy-resolving ability of the model is important to adequately simulate heat and salt transfer in the sea interior and obtain a correct circulation pattern. To ensure model stability without excessively damping physical mode of its solution a parameterization of lateral viscosity has been implemented based on a bi-harmonic operator with Smagorinsky coefficient $C = 3$ as discussed in Griffies and Hallberg (2000). Lateral diffusivity is parameterized with a simple Laplacian scheme with constant coefficient $A_h = 1$. Heat and salt advection is approximated using a total variation diminishing scheme from Sweby (1984), which acts as a second-order scheme in most cases, except high-gradient frontal zones. In the vertical a grid is set with rather fine resolution varying from 2 m in the upper sea layer to 30 m in deep waters. This minimizes numerical errors in advection terms and prevents excessive accumulation of the errors in the long-term. Vertical viscosity and diffusivity are parameterized via a scheme based on Richardson number (Munk and Anderson, 1948) with variable coefficients: $K_m = (10^{-5} - 10^{-3})$ m$^2$ s$^{-1}$ for viscosity and $K_h = (10^{-7} - 3*10^{-4})$ m$^2$ s$^{-1}$ for diffusivity and thermal conductivity. Model time step is 5 min.

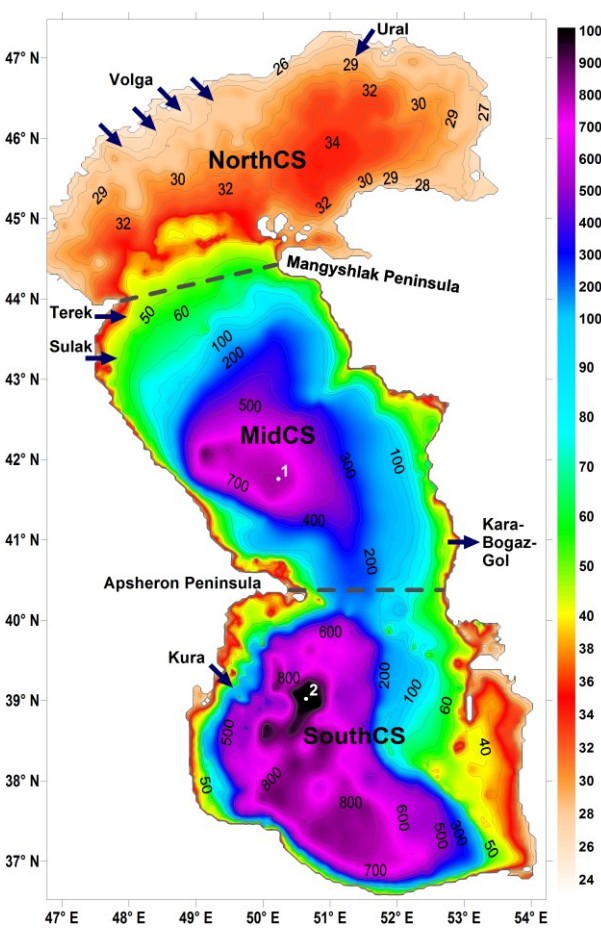

**Fig. 1.** Caspian Sea bathymetry, used in the model (depths relative MSL, m). The average sea surface height and model vertical grid origin are 28 m. Dashed lines indicate conventional separation of the sea into three basins: northern (NorthCS), middle (MidCS) and southern Caspian (SouthCS). Arrows designate water inflows due to rivers accounted in the model and the sea water outflow into the Kara-Bogaz-Gol Bay. Numbers 1 and 2 indicate locations of the deep-water stations from Tuzhilkin and Kosarev (2004), for which reference observational T and S data will be given further.

## 2.2 External forcing

Monthly mean river runoff data were used to prescribe the discharge of the Volga, Ural, Kura, Terek and Sulak Rivers. The outflow into the Kara-Bogaz-Gol Bay was set using annual mean data. Atmospheric forcing was prescribed using the ECMWF Era-40 atmospheric reanalysis dataset (Kallberg et al., 2004), chosen for several reasons. First, the data cover an extended period (from 1957 to 2002) comprising one of the most vivid episodes of global climate change – the climatic regime shift of 1978. This allows investigating the Caspian Sea response to such global events. The other advantage of the Era-40 reanalysis is its relatively high spatial resolution (1.125°), which is still rather coarse for the Caspian Sea with dimensions 8°×11°, but is sufficient to resolve main features of the atmospheric circulation in the region, as has been shown by Ibrayev et al. (2010). The Era-40 temporal resolution of 6 hours allows simulating the diurnal air-sea interaction cycle and the synoptic variability mode. As any global reanalysis product, Era-40 has errors, specific for a particular region of the

planet (Berg et al., 2012; Cattiaux et al., 2013). Therefore, we have partially corrected the Era-40 wind and precipitation fields for better consistency with the available climatology atlases of the Caspian region (Panin, 1987; Terziev et al., 1992): wind speed was increased by 15%, precipitation was decreased by 30%. The performed corrections as well as the model sensitivity to them are considered in detail in Dyakonov and Ibrayev (2018). The prescribed atmospheric parameters together with the parameters of the sea surface, obtained in the model, are used to compute air-sea fluxes based on the approach of Launiainen and Vihma (1990): evaporation, sensible and latent heat fluxes and the momentum flux. Precipitation and radiative heat fluxes are taken directly from Era-40. The fluxes are dynamically amended due to sea ice cover, simulated in the submodel of sea ice thermodynamics, described in Schrum and Backhaus (1999).

## 2.3 Initial conditions and model "spin-up"

The model was initialized with the climatic mean 3D-fields of temperature and salinity for January (Kosarev and Tuzhilkin, 1995). These fields have been considerably smoothed and averaged over an extended period of time, and, therefore, lack realistic cross-shore gradients and many other details, particularly in shallow regions. While the distribution of temperature in such areas adjusts rather quickly due to atmospheric impact, the salinity field is a lot more passive and requires an additional "spin-up" model run: the model is run for 5 years with a relaxation of sea surface salinity (SSS) in the southern Caspian basin. This is necessary to avoid excessive growth of salinity in the upper layer of the basin, until a fresh-water anomaly, associated mainly with the Volga River's runoff, appears along the western coast of the Mid Caspian. It is this anomaly that supplies relatively fresh water to the south in an amount sufficient to compensate intense evaporation. After 5 years a realistic salinity distribution in the Mid Caspian is achieved, and the SSS relaxation in the southern Caspian is no longer required to balance the salt budget of this basin. The resultant salinity field is then used as the initial condition for the main model run, discussed in the following sections.

## 3 External forcing variability

Figure 2 shows the evolution of external forcing components for the period considered. The Caspian Sea water budget is a sum of river discharge (~300 km$^3$ year$^{-1}$) and precipitation (~100 km$^3$ year$^{-1}$) approximately balanced by evaporation (~400 km$^3$ year$^{-1}$) and the outflow into the Kara-Bogaz-Gol Bay (~30 km$^3$ year$^{-1}$) (Terziev et al., 1992). The underground water contribution is thought to be insignificant (~4 km$^3$ year$^{-1}$) (Zektser et al., 1984). Evaporation is the only component that cannot be directly measured, and therefore it is computed by the model based on air and sea surface parameters. The evolution of the net input of the other three water budget components as well as river discharge and precipitation are separately presented in Figures 2a, 2b and 2c respectively. In the late 1970s one can note a sharp rise (~20 %) in the net water input, which was a consequence of the climatic regime shift, mentioned earlier. The shift was also associated with an increase of air humidity in the Caspian region, followed by a trend change in the evolution of radiative fluxes: both solar and thermal radiation (absolute value) intensities imply warming after 1980. We will refrain from discussing the reasons and mechanisms

140 of such abrupt variations and merely ascertain the fact, that the data, used to prescribe the external forcing in the model, contain the signal, associated with the climatic shift of 1978. Notably, there is no significant long-term change in the average air temperature and wind speed module present in the data (Figs. 2d and 2e).

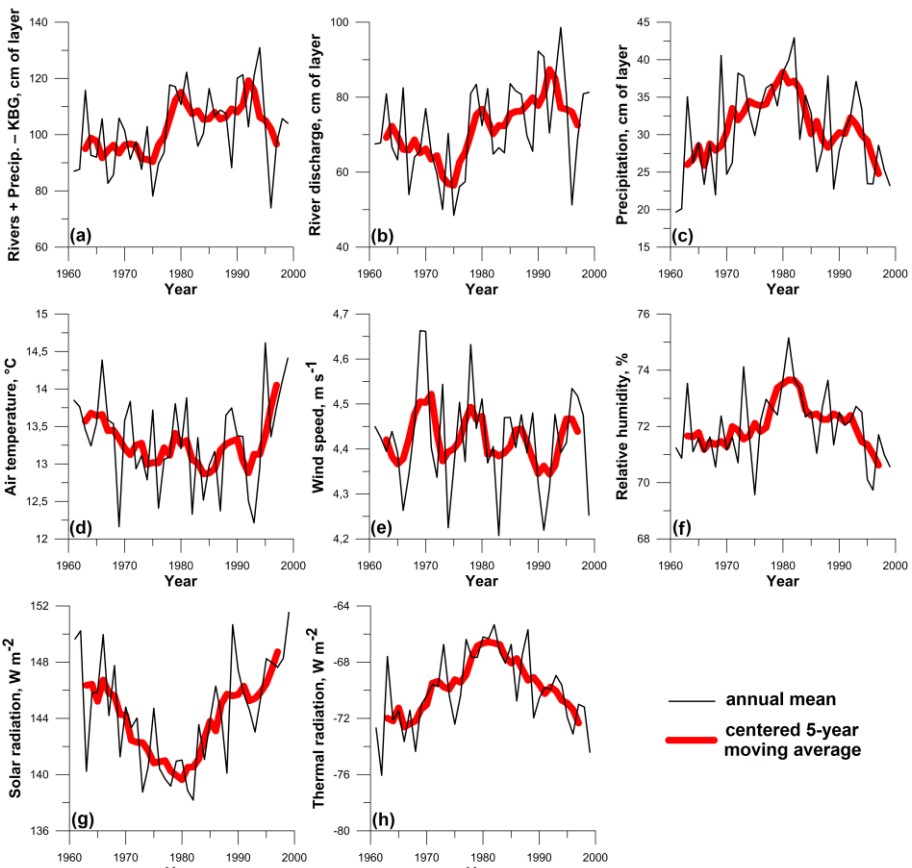

Fig. 2. Long-term variability of the forcing components: (a) – sum of riverine water input and precipitation with the deduction of
145 the outward flux into the Kara-Bogaz-Gol Bay; (b) – riverine water input alone; (c) – precipitation; (d) – air temperature (°C); (e)
– wind speed module (m s$^{-1}$); (f) – relative humidity (%); (g) – surface solar radiation (W m$^{-2}$); (h) – surface thermal radiation (W
m$^{-2}$). All atmospheric parameters and fluxes were averaged over the sea area; water fluxes (a, b, c) are given in terms of the
corresponding sea level increment.

## 4 Surface circulation

150 We start discussion of the model results with a brief review of surface circulation patterns. This will help to shed some light on further results, as it is the circulation in the upper active layer that mostly determines physical processes occurring in the entire water column. Figure 3 shows monthly mean sea surface currents (SSC) in January and July, averaged over 1961-1977 (before the regime shift of 1978) and 1978-2001 (after the shift). This division allows assessing how the climatic shift influenced the SSC field. July pattern altered insignificantly, so only the plot for the first period is presented. Winter
155 circulation, on the contrary, altered rather noticeably, but only in the MidCS basin, where the direction of the open sea main

flow changed by 45° counterclockwise. We provide these fields only for reference and will not further investigate their variability, as the impact of the climate shift on the SSC is beyond the scope of the present study.

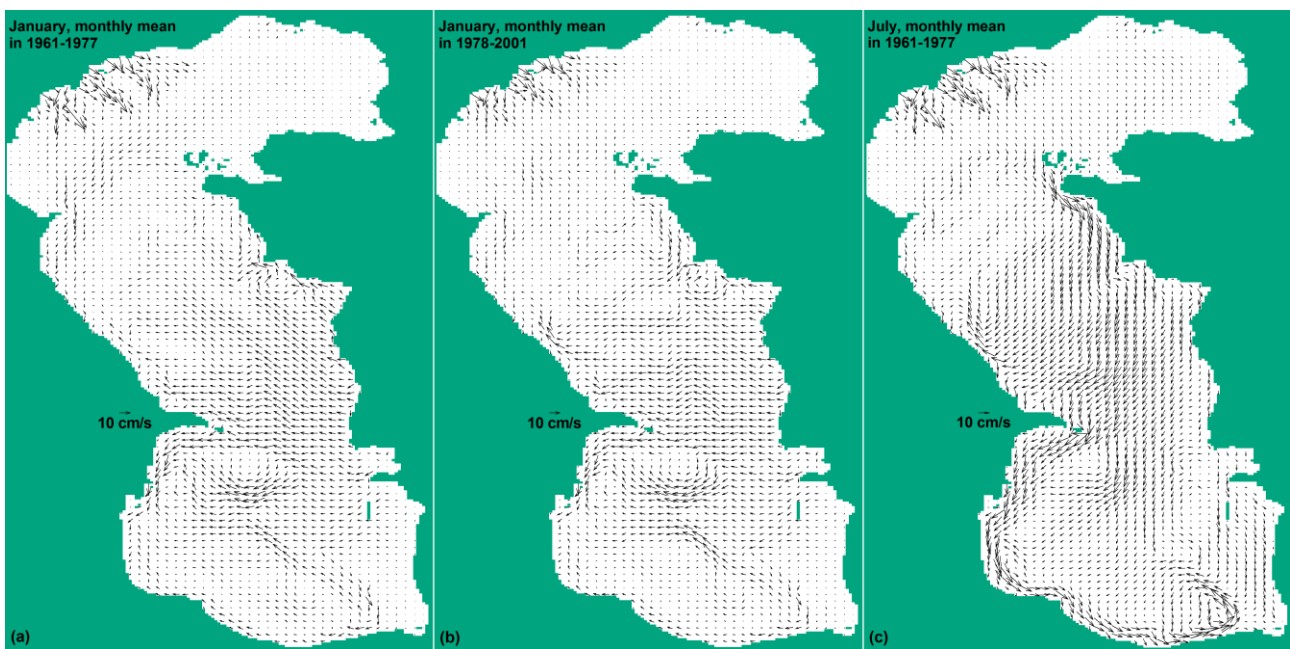

**Fig. 3. Model monthly mean surface currents in January (a and b) and July (c) averaged over 1961-1977 (a and c) and 1978-2001 (b).**

Typical distributions of sea surface salinity (SSS) and temperature (SST) are shown in Fig. 4. Unlike currents in Fig. 3, these are instantaneous fields, though clearly correlating with many SSC features. Figure 4a vividly demonstrates the differences in thermal regime of the three Caspian basins in late winter: while SST in NorthCS is around zero (the basin is covered by ice sheet), SouthCS waters are much warmer (up to 11°C). MidCS basin is subject to intrusions from both north (cold elongated current propagating along the western shore) and south (warm anomaly in the open sea). The most distinctive feature in SST field during summer is a cold anomaly in the eastern part of MidCS (Fig. 4b) created by an upwelling, which occurs along the eastern shore due to northwest wind typical for this region in summer. The same wind accounts for a large fresh-water intrusion from NorthCS into MidCS (Fig. 4c), which is formed by a relatively strong jet current near the Mangyshlak Peninsula (see Fig. 3c). Although existence of this jet is consistent with satellite imagery (Kostianoy et al., 2013), the intensity of the corresponding fresh-water transport is evidently overestimated by the model, as these regularly occurring intrusions decrease average SSS in MidCS below that observed by ~0.5 psu. This is likely due to excessive numerical viscosity of the tracer advection scheme implemented in the model. Figure 4d shows a reverse situation, characterized by an intrusion of relatively salty MidCS waters entering NorthCS and an opposite process occurring along the western MidCS shore. Similar intrusions are noted between MidCS and SouthCS basins (Figs. 4c, 4d). Thus, exchange of water masses with contrast salinity between Caspian basins plays an important role in formation of the thermohaline regime of the sea.

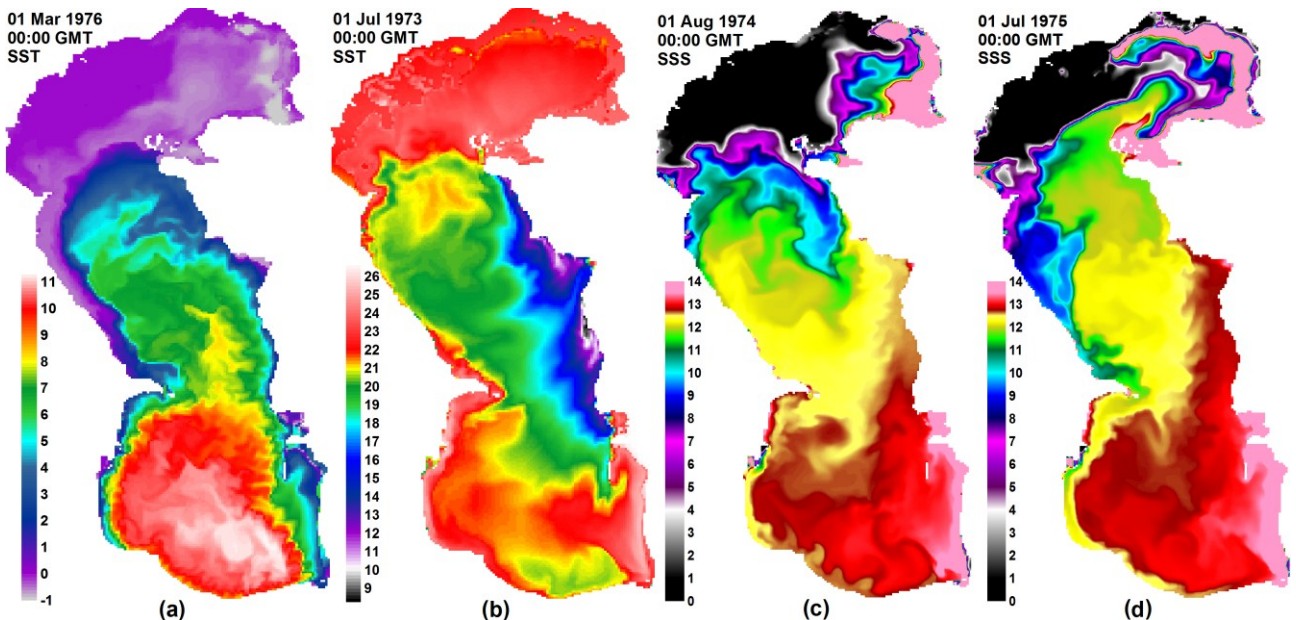

Fig. 4. Model instantaneous sea surface temperature (SST) and salinity (SSS): (a) SST (°C) on March 1, 1976; (b) SST (°C) on July 1, 1973; (c) SSS (psu) on August 1, 1974; (d) SSS (psu) on July 1, 1975.

## 5 Model validation

To assess the magnitude of model errors we will compare the evolution of its solution with in-situ observations. First let us consider the reconstructed sea surface height (CSSH), which is an integral indicator of the model quality, as it depends on the sea surface temperatures that reflect thermohaline circulation of the entire sea. Figure 5 compares the observed sea level in the vicinity of Baku (Apsheron Peninsula) with that obtained by the model. Until the sharp decline of 1975 there is a good match of the two curves, which indicates a correct description of sea water balance components. Yet sharp changes of the sea level are not well reproduced in the following period, possibly due to errors in the model and/or inaccuracies in the external forcing data, which led to a considerable discrepancy (up to 35 cm). As a result, the sea surface area is overestimated by the model, and, in turn, so is the net evaporation flux, which is why the model CSSH has a slow downtrend relative observations and matches them again in 1992. This negative feedback between the Caspian Sea level and its surface area was shown to be significant in the earlier work by Dyakonov and Ibrayev (2018). In an auxiliary experiment accurate water balance was demonstrated, when the model was started from 1978 with the correct initial CSSH, which suggests that errors in water budget components occur only in the mid-1970s, i.e. when first changes in the regime were detected. Overall, the evolution of the Caspian Sea surface height is reconstructed reasonably well, and this fact alone refutes all of the hypotheses relating its rapid rise in 1978-1995 with various earlier speculations in literature such as changes of the seabed, underground infiltration of the Aral Sea waters into the Caspian, variations of underground riverine discharge and other factors that were not taken

into account in the model. Indeed, our results are consistent with the theory on the dominant role of the global climate fluctuations in the Caspian Sea level variability on a decadal timescale (Frolov, 2003; Panin and Diansky, 2014). Thus, the sharp level growth was caused by the above mentioned climate regime shift of 1978, and the corresponding signal is present in the forcing data we use (Figs. 2b, 2f).

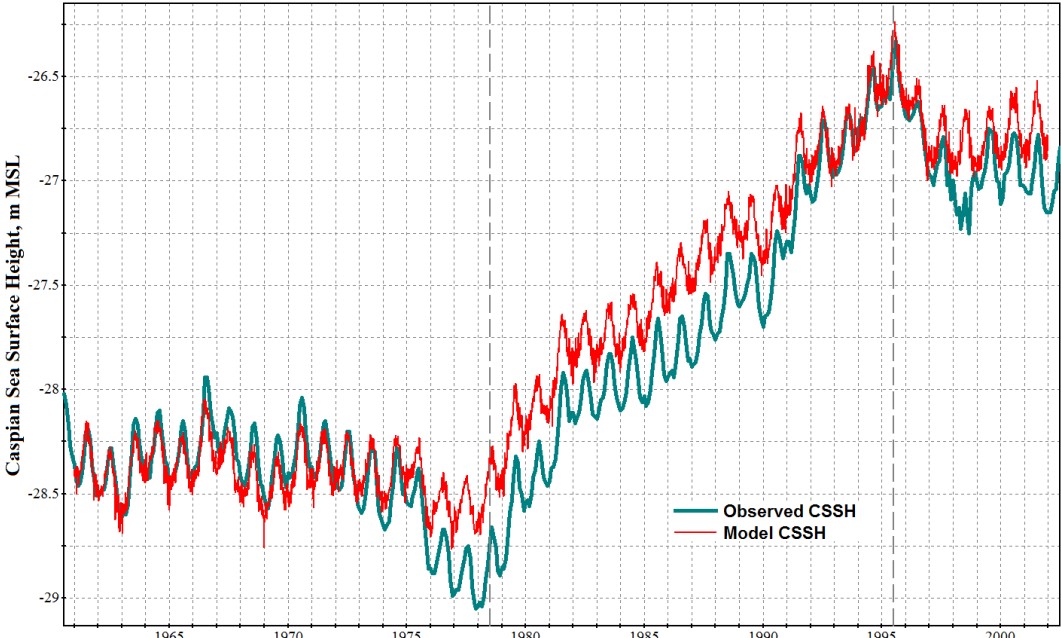

**Fig. 5. Caspian Sea surface height (CSSH) in the vicinity of Baku: observations and model reconstruction (m, MSL).**

To analyse deep water properties we use the long-term measurements of T and S at two particular points located in the central parts of the middle and southern Caspian basins (locations 1 and 2 on Fig. 1) from Tuzhilkin and Kosarev (2004), who have studied the evolution of temperature and salinity in deep-water zones of the Caspian Sea in 1956-2000. This

interval was divided into four distinct periods: (1) quasi-stationary conditions in 1956-1967, (2) harsh winters and low river influx in 1968-1977, (3) increased river influx in 1978-1995 and (4) regime saturation in 1995-2000. Unfortunately, we do not have the source data available, so only the mean T and S values for these four periods will be used in each location. Figures 6 and 7 compare these averaged values at 100 m depth with those obtained by the model. Overall, one can note a correlation of the reconstructed evolution with observations, more so in SouthCS than in MidCS. Significant discrepancy in

salinity at location 1 is caused by the aforementioned SSS error in MidCS. As a result, salinity stratification of the upper active layer is overestimated in this basin, which obstructs intense deep convection responsible for the observed increase of salinity at 100 m in 1968-1977 (Fig. 6), caused by harsh winter conditions of the period. In SouthCS (location 2) model salinity is much closer to observations with the exception for the first period, apparently due to inadequate initial conditions, corresponding to the climate mean rather than instantaneous values of 1961. However in three years model salinity reaches

values close to the mean observed ones. Systematic overestimation of temperature by 0.5-1.5 °C at both locations reflects the errors in the description of vertical mixing including insufficient convection intensity and can be considered the model error.

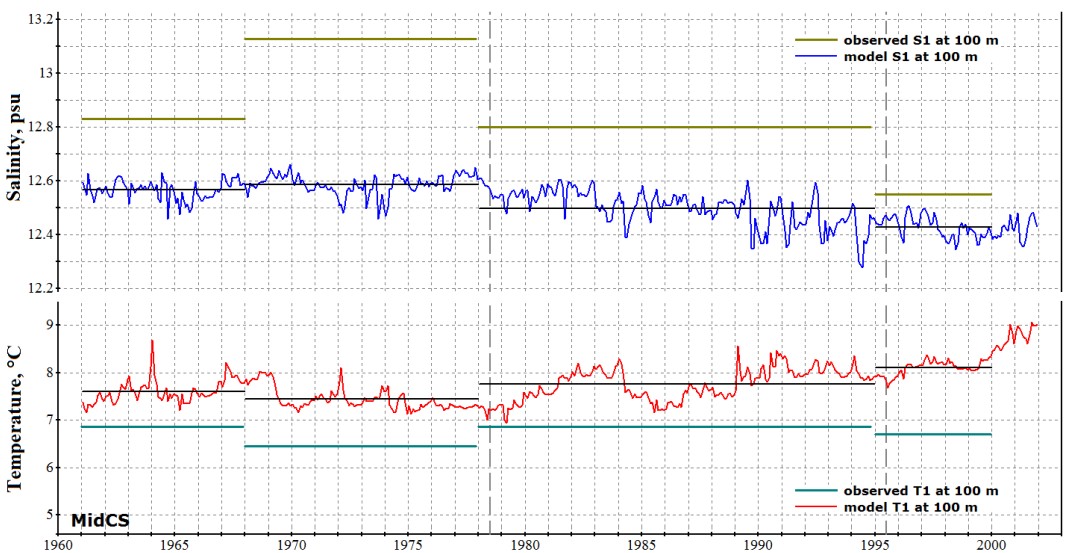

Fig. 6. Salinity (upper panel) and temperature (lower panel) at 100 m depth at location 1 (MidCS, see Fig. 1) obtained in the model
and those observed. The observational data are plotted as a mean value, constant for four different periods. Mean model values for the corresponding periods are shown in thin black lines.

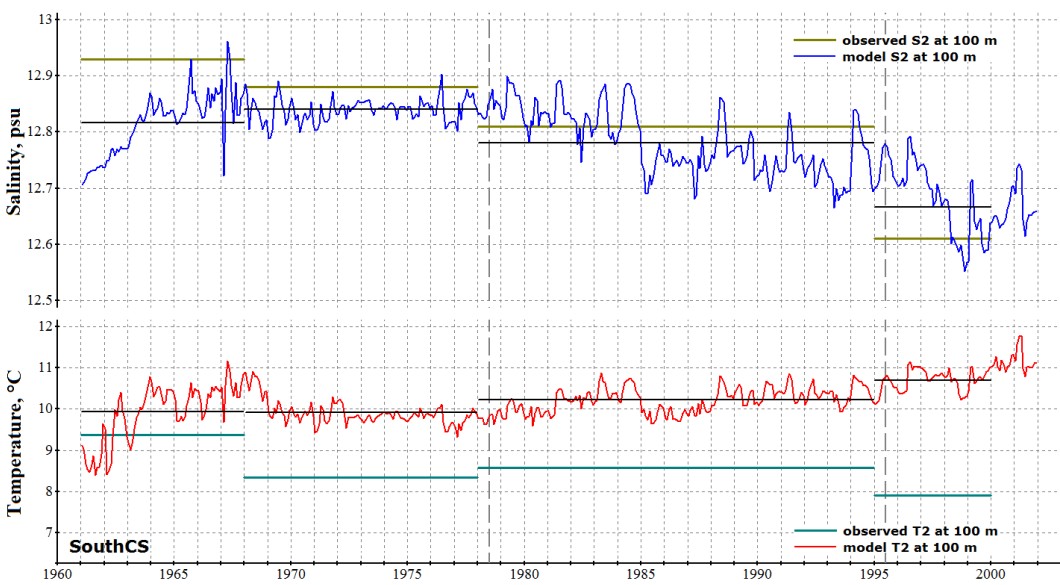

Fig. 7. Same as Fig. 6 for location 2 (SouthCS, see Fig. 1).

## 6 Long-term trends of thermohaline properties

### 6.1 Northern Caspian

The northern Caspian is a very shallow estuary of the Volga and Ural rivers, the circulation of its waters is strongly influenced by their discharge and wind. Due to the shallow depth (4-5 m in most of its area) the water column is almost always well-mixed throughout the year, allowing us to analyze surface properties only. Figure 8 shows the evolution of sea surface salinity (SSS) in all of the three basins. The amplitude of the SSS annual oscillations increases northward, which is a direct consequence of the riverine run-off distribution in space. As one can see from the Fig. 8, SSS in the northern basin fluctuates around 8 psu until the climate regime shift of 1978 and then reaches a new quasi-equilibrium state with annual mean value slightly below 7 psu. The time required for this transition period is rather small and amounts to 3-4 years. After 1980 the SSS trend stabilizes, but an additional drop down to 6 psu occurs in the 1990s. In the other two Caspian basins SSS trends are similar, but their rates are smaller by an order of magnitude. Overall, SSS evolution in the entire sea correlates with river discharge and air humidity, and the results presented here are consistent with the observations (Tuzhilkin et al., 2011).

Notably, the reconstructed evolution of the NorthCS salinity field is rather sensitive to model design, particularly to bottom drag parameterization. An important feature of this basin is that it serves as a transit zone for fresh riverine waters, moving into the MidCS and SouthCS basins to be evaporated there. This leads to a continuous loss of the net mass of salt in the northern basin, which can be compensated only by recurrent intrusions of the MidCS saline waters induced by wind, as shown in Fig. 4d. However, such intrusions are usually brief, so the amount of salt, that enters and remains in the northern basin, greatly depends on the bottom drag resistance to the currents transporting it. Use of a too viscous bottom drag parameterization in our prior experiments caused a gradual decline of mean NorthCS salinity down to zero within a decade. Therefore, a new parameterization scheme, that is more adequate for such shallow regions, has been devised, which allowed to stabilize the salinity evolution here at a level close to that observed. Nonetheless, salinity distribution in NorthCS is still somewhat inaccurate as compared to observations: the fresh-water tongue, associated with the Volga River, extends southward too far, often shifting salinity gradient maximum close to MidCS waters (see Fig. 4c). In these conditions wind drives very fresh water masses into the MidCS basin, decreasing its surface salinity down to ~12 psu on the average (Fig. 8), which is about 0.5-0.7 psu lower, than observed.

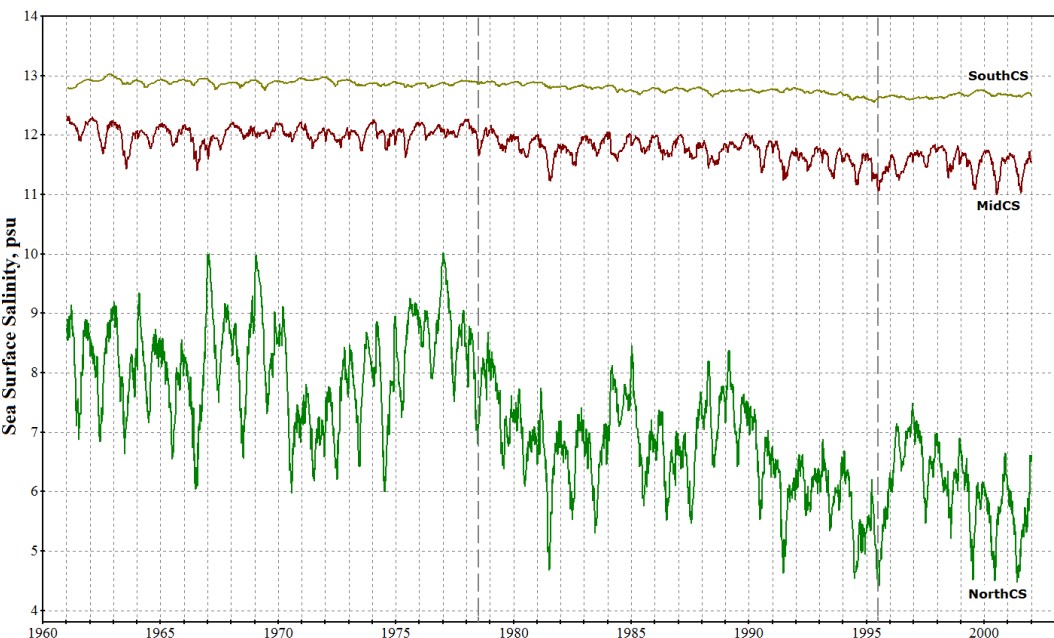

**Fig. 8. Evolution of sea surface salinity (SSS) averaged over three Caspian basins (psu).**

### 6.2 Middle Caspian

The middle and southern Caspian basins have maximum depths of 800 m and 1000 m respectively, so vertical mixing
processes play a much greater role in thermohaline circulation here than in the north. In MidCS autumn-winter convection is
thought to create a mixed layer of depth 200 m and to mix the entire water column during the coldest winters, e.g. in the
winter of 1969 (Terziev et al., 1992). However, more recent papers (Tuzhilkin and Goncharov, 2008; Tuzhilkin et al., 2011)
suggest, that throughout the period considered, convective mixing occurred only in the upper 100 m layer and did not reach
the Caspian abyssal waters even in the most severe winters. Our results support these conclusions: in the numerical
reconstruction average depth of winter convection in the deep parts of the MidCS basin is about 80 m. The above mentioned
0.5-0.7 psu underestimation of MidCS surface salinity significantly decreases convection intensity, but auxiliary sensitivity
experiments have shown that convection depth does not exceed 110-120 m, even when this error in the SSS field is
artificially compensated.

Figures 9, 10 and 11 show the reconstructed evolution of salinity, temperature and density at different depths in the MidCS
basin. At a depth of 250 m the effects of convective mixing are noted only in the coldest winters and are absent at 500 m and
below (Fig. 10). In the active layer (upper 100-150 m) the thermohaline properties exhibit a clear seasonal cycle and have no
long-term trend until 1978, which indicates a quasi-stationary circulation regime. After the climate shift of 1978 the upper
layer salinity begins a gradual decline (Fig. 9), associated with intensification of river discharge and increase of air humidity
in the Caspian region. These downtrends cease only after the next climatic shift in the mid-1990s, when a new quasi-
stationary sea circulation regime is achieved. Because the freshening signal, associated with the first shift, originates at the

surface, the rate of the salinity downtrend decreases with depth, which strengthens sea stratification and diminishes convection-driven ventilation of deep waters. These results are qualitatively consistent with observations (Tuzhilkin and Kosarev, 2004; Tuzhilkin et al., 2011). The weakening of convection also accounts for the reduction of winter SST, noted after the shift of 1978 in MidCS, and for the upward trend in subsurface temperatures (Fig. 10), as was suggested in Tuzhilkin et al. (2011).

At greater depths (500 m and deeper) the influence of changes in external conditions becomes almost indistinguishable from the accumulating model errors, that account for a slow downtrend (~0.1 psu / 40 years) in the average salinity and an uptrend (~1° C / 40 years) in the average temperature. These trends are caused by advective and diffusive mixing and are inevitable in presence on small but non-zero T and S vertical gradients. According to Tuzhilkin and Goncharov (2008), the only process that can counteract it, is down-slope cascading – slow sinking of cold saline waters along the slope of the northern and eastern continental shelves. Despite its important role, this process is not fully taken into account by the model, which is why it yields these erroneous slow trends. The reason is that at depths greater than 30 m the model uses z-coordinate grid, and bottom slope is represented as a set of horizontal stairs obstructing cascading process. To overcome this z-coordinate deficiency a parameterization of cascading should be implemented in the model. In the active layer, on the contrary, the model errors do not conceal the actual variability of water properties: the long-term trends alternate with quasi-stationary circulation regimes in correlation with external forcing variations.

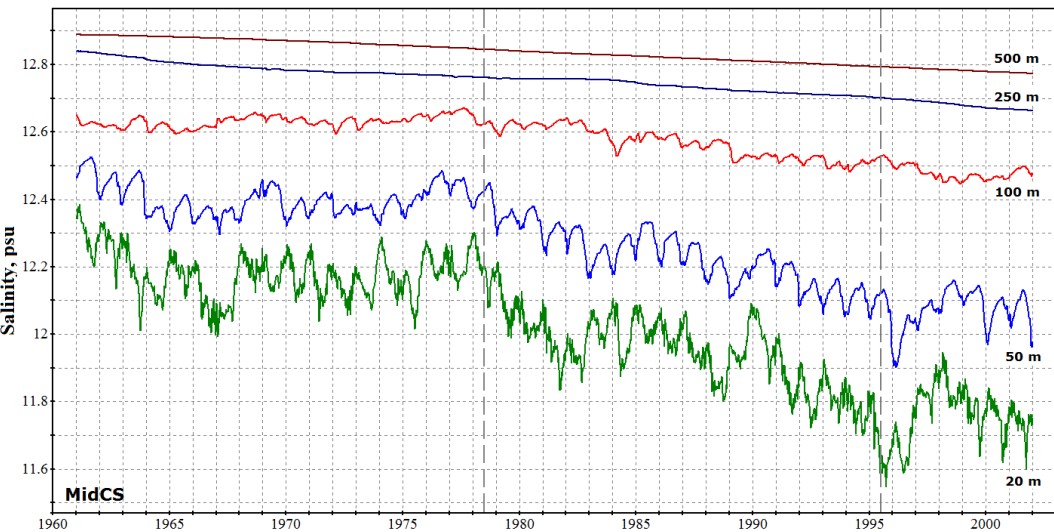

**Fig. 9. Evolution of salinity (psu) at different depths averaged over the MidCS basin.**

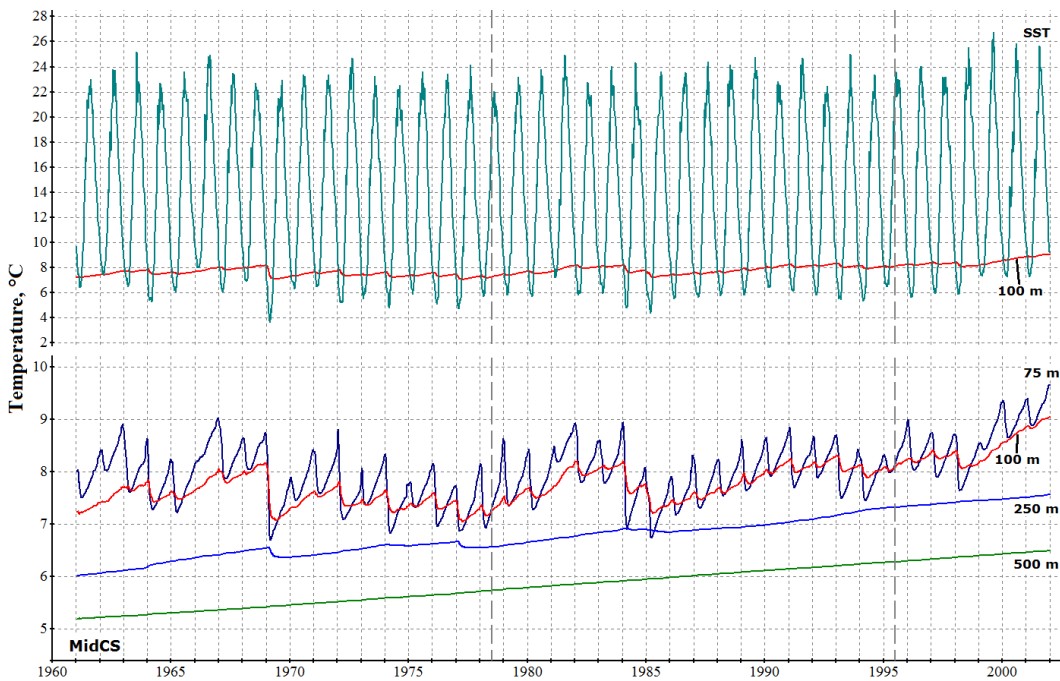

**Fig. 10. Evolution of temperature at different depths (SST – sea surface temperature) averaged over the MidCS basin (°C).**

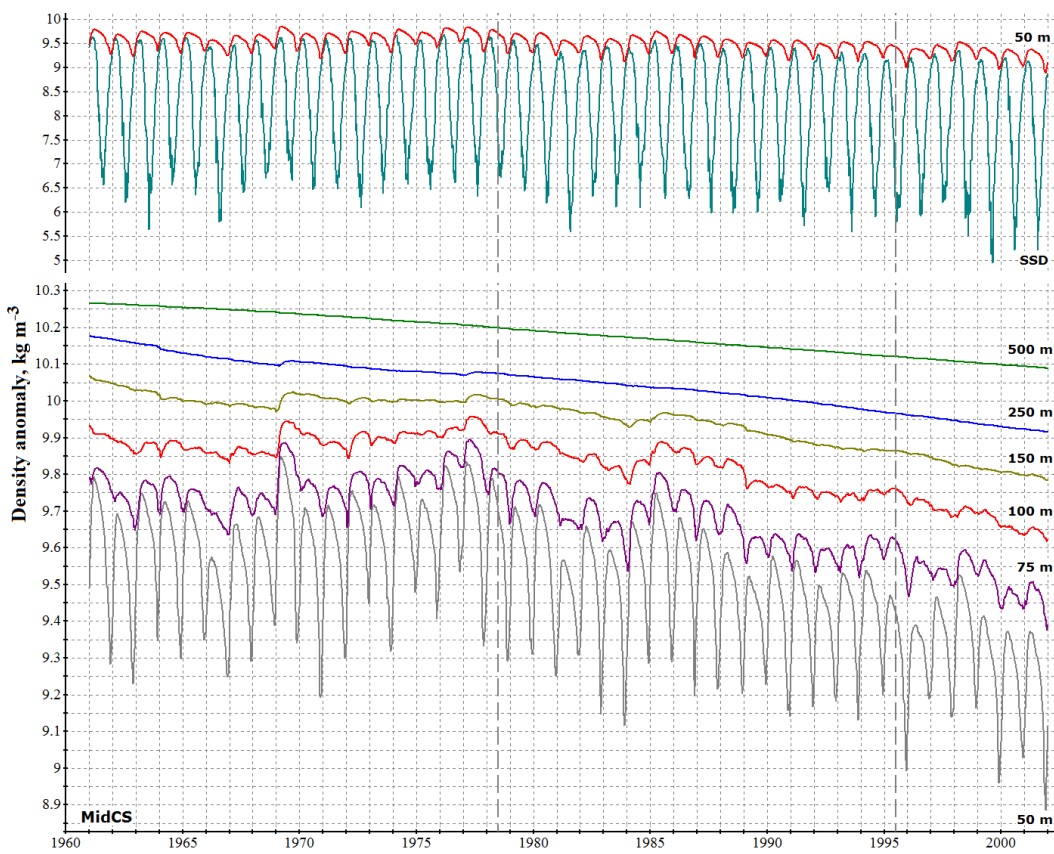

**Fig. 11. Evolution of density anomaly (kg m⁻³) at different depths (SSD – sea surface density) averaged over the MidCS basin.**

### 6.3 Southern Caspian

The reconstructed salinity, temperature and density in the SouthCS basin are presented in Figures 12, 13 and 14. The southern Caspian basin is the most distant one from the Volga River's mouth and has the strongest evaporation throughout most of the year (Panin, 1987), therefore salinity field in its active layer is rather sensitive to water exchange with relatively less salty MidCS basin. In order to attain a circulation regime that would balance the salt budget of SouthCS, a 5-year spin-up model run with SSS relaxation was necessary, as we have described in section 2.3. However, after the SSS field had been

released, it took three more model years to reach quasi-equilibrium circulation in the upper 100 m sea layer (Figs. 12, 13). During the first two years surface salinity grows rapidly, which leads to an intensification of convection-driven mixing in the active layer during the third year of the run with relatively sharp rises of temperature and salinity at the depth of 100 m. By the fourth year of the run (in 1964) a vertically quasi-homogeneous salinity distribution is achieved (Fig. 12), characterized by a slight positive deviation (~0.1 psu) in the active layer from the mean climatic values from Kosarev and Tuzhilkin

(1995). As a result, the maximum convection depth exceeds that observed by 10-15 m: according to Terziev et al., 1992 convective mixing processes in SouthCS span the upper 70-80 m layer and reach 100 m only in its northern part. In the model reconstruction lower boundary of convective mixed layer is located at the depth of 80-90 m in the central area of the

basin. Thus average temperatures at 100-150 m are overestimated by 1–2° C due to overly intense mixing with warmer surface waters during winter.

After the first four years of the model run a steady circulation regime is achieved in the upper 100 m layer, which persists until the 1980s. The impact of the climatic shift of 1978 on thermohaline properties in SouthCS is similar to that obtained in MidCS, but has a 3-year time lag, required to adjust MidCS circulation to the forcing variation. In 1981 a transition begins to a new circulation regime, characterized by a restoration of stable salinity stratification and an additional increase of temperatures in the lower part of the active layer. Thereby autumn-winter convection in SouthCS weakens, as can be clearly

seen in the evolution of density at 75 m on Fig. 14. Like in the middle Caspian, slow trends in temperature and salinity below 250-300 m in SouthCS are a result of vertical advective and diffusive mixing in absence of sufficient deep water ventilation via downsloping cascading from the eastern shelf. At the depth of 250 m the effects of the second climatic shift are still observed, and this is the maximum depth, to which a signal of external forcing variability propagates, both in MidCS and SouthCS basins.

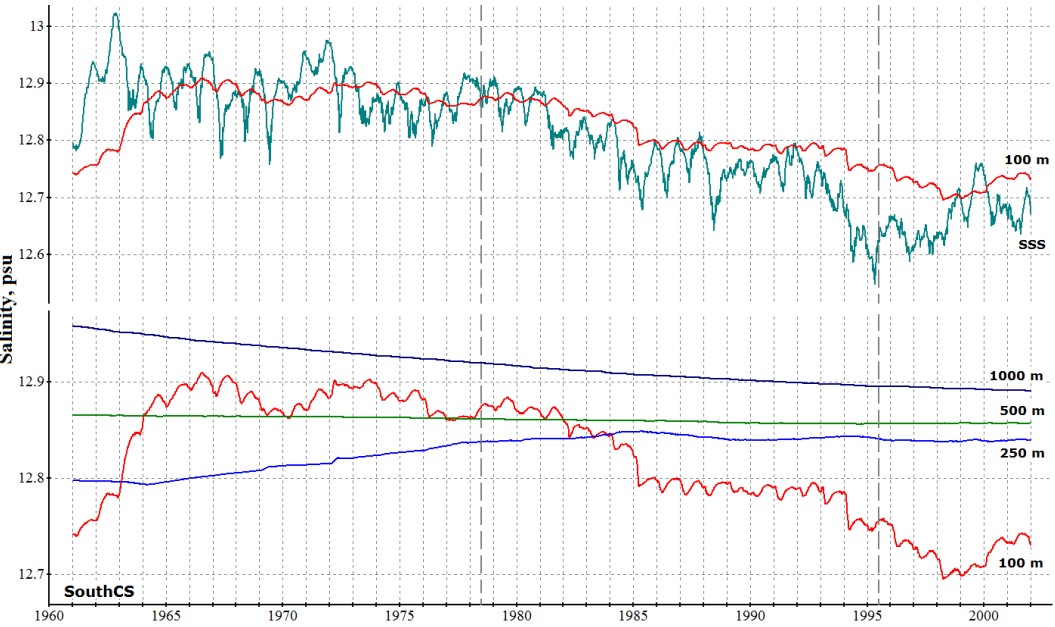

**Fig. 12. Evolution of salinity (psu) at different depths averaged over the SouthCS basin.**

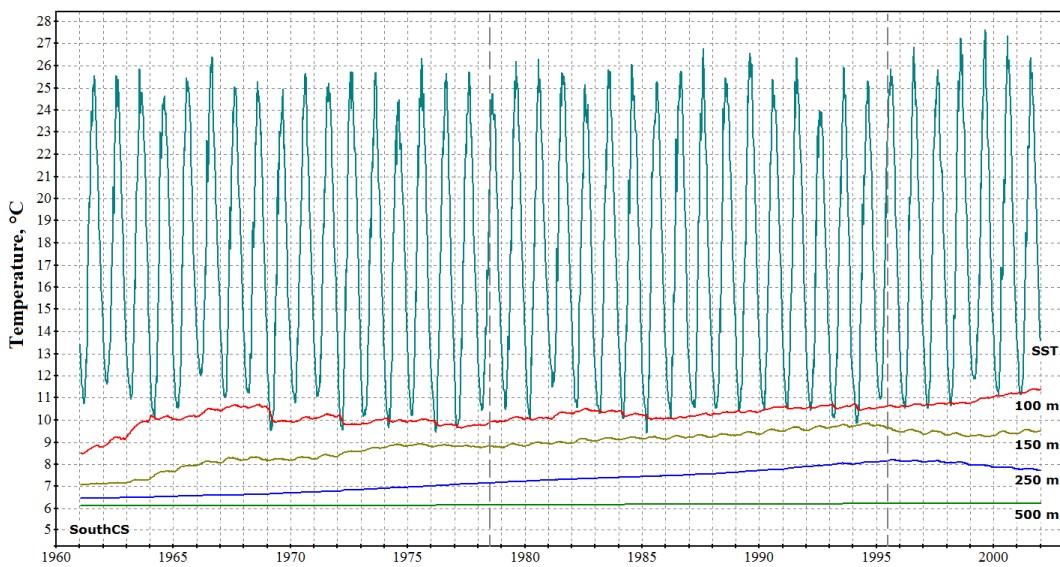

**Fig. 13. Evolution of temperature at different depths (SST – sea surface temperature) averaged over the SouthCS basin (°C).**

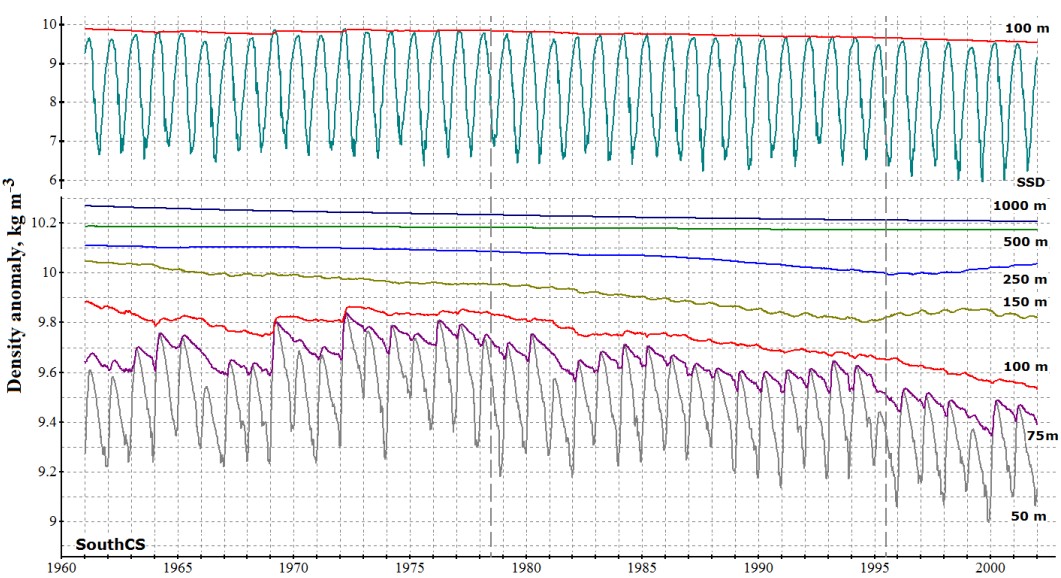

**Fig. 14. Evolution of density anomaly (kg m⁻³) at different depths (SSD – sea surface density) averaged over the SouthCS basin.**

## 7 Summary and conclusions

We have considered a long-term numerical reconstruction of the Caspian Sea thermohaline circulation in 1961-2001. The model reproduced a quasi-stationary regime that lasted until 1978 and, at least qualitatively, the sea response to the global climate shift that had occurred in 1976-1978. The influence of surface circulation on the thermohaline regime of the sea has been discussed, a crucial role of the exchange of waters with contrast parameters between the three Caspian basins has been demonstrated. A correct reconstruction of the water balance in 1978-1995, i.e. during the period of a rapid sea level rise (~2.5

m), confirms that the level rise was associated with the variability of riverine and atmospheric forcing, rather than other factors that are not accounted for by the model. Thus, our results are consistent with the commonly recognized theory, relating the Caspian Sea level fluctuations with global climate changes.

During the first 15-17 years of the experiment a quasi-stationary circulation pattern was obtained with a clear seasonal cycle and almost no long-term trends in the evolution of temperature and salinity in the active layer. Due to model errors in reproducing surface salinity field the depth of winter convection in the middle Caspian is about half of that estimated in Terziev et al. (1992), although in better agreement with the results of more recent studies (Tuzhilkin and Goncharov, 2008; Tuzhilkin et al., 2011). At greater depths, below the active layer, slow trends in the evolution of thermohaline properties were

obtained as a result of insufficient ventilation of these waters. The reason is that the model does not fully take into account down-slope cascading processes because of using z-coordinate. The error accumulation rate amounts to ~1° C / 40 years for temperature and ~0.1 psu / 40 years for salinity. At the intermediate depths (200-300 m) both these trends and the effects of external forcing variability are noted, while below 250-300 m the latter are absent.

After 1978 the non-trend circulation mode was replaced by a transition to a new circulation regime due to a shift that had

345 occurred in the global climate. This transition was associated with downtrends in salinity field, which led to strengthening of density stratification in the upper sea layers and weakening of autumn-winter convection. As a result of an increased isolation from surface waters during winter, the temperature at 100-200 m showed an uptrend. The surface salinity in the northern and middle Caspian responded to the increased river discharge almost simultaneously, while the corresponding trend in the southern Caspian SSS occurred with a 3-year time lag, which indicates a much greater interdependence of the middle

Caspian with the northern basin rather than the southern one. Overall, the reproduced sea response to the climatic shift of 1978 is discernible despite considerable model errors and it is consistent with the observational data analysis, presented in Tuzhilkin et al. (2011). The next climatic shift of 1995 stabilized the salinity trends, and a new circulation regime was achieved.

When modeling the Caspian Sea circulation, the greatest challenge is to keep salinity distribution in the active layer close to

355 that observed. Even slight errors in the salinity field significantly modulate intensity and depth of convective mixing and, consequently, alter thermohaline circulation patterns of the entire sea. Two major factors determine the deviations of salinity: external forcing errors and model quality, particularly the description of inter-basin water mass exchange, as the three Caspian basins have different salinity regimes. Correct simulation of deep water properties requires taking into account down-slope cascading, which is an important mechanism of ventilation and renewal of the abyssal Caspian waters. However,

despite all of its errors and simplifications, the model qualitatively reproduced the evolution of the Caspian Sea thermohaline circulation and its response to external forcing variations.

## Author contributions

The research was carried out by GD under the supervision of RI.

## Competing interests

The authors declare that they have no conflict of interest.

## Acknowledgements

The work was carried out at the Northern Water Problems Institute of the Karelian Research Center with the financial support of Russian Science Foundation grant no. 14-17-00740 "Lakes of Russia: Diagnosis and Prediction of State of Ecosystem under Climate Changes and Anthropogenic Impacts." The research was carried out using the equipment of the shared research facilities of HPC computing resources at Lomonosov Moscow State University (Sadovnichy et al., 2013) and Joint Supercomputer Center of the Russian Academy of Sciences.

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
