# Peer review of "Long-Term Evolution of the Caspian Sea Thermohaline Properties Reconstructed in an Eddy-Resolving OGCM"

_Ocean Science, 2018_

## Referee Comment (RC1) · Anonymous Referee #1 · 5 Dec 2018

The authors have reconstructed the thermohaline properties of the Caspian Sea for a period of 40 years (1961-2001) using a numerical model. The model results are then used to study the impact of external forcing on the evolution of salinity, temperature and density at different depths of the Nothern, Middle and Southern Caspian Sea. One of the interesting features is the impact of the climate shift in 1976-1978 on the thermohaline properties. The paper also discusses the challenges of long term modeling of the Caspian Sea. These include accumulating model errors caused by advective and diffusive mixing and bottom drag parameterization in the shallow areas of Northern Caspian Sea.

[Figure]

To my knowledge, this is the first study to model long-term (40 years) variability of the Caspian Sea. Therefore, it contributes to a better understanding of the impact of climate shift on the physical properties of the area. It is also a good reference for numerical modelers that aim at reconstructing long-term variability of enclosed water bodies, as it points out the difficulties of modeling such environments. Hence, I propose that the paper is accepted with minor revisions:

1. It is mentioned that the downsloping cascading process of cold saline waters along the slope of nothern and eastern shelfs are not fully taken into account by the model. Is this due to using z-level grids in the model? If not, it would be interesting to know how this process can be better resolved in numerical models.

2. The model is validated against sea surface height at Baku and is shown to be able to well reproduce its long-term variability. However, for the salinity and temperature evolution, the comparison against observed data has not been shown (Although it is mentioned that the model is in good agreement with observations). It would be interesting to have a more detailed comparison of the model thermohaline structure against observations shown in Tuzhilkin and Kosarev, 2004. My understanding is that the model shows more salinity stratification in the period of 1960-1978 in the middle Caspian Sea. However, it is hard to reach a conclusion because the results shown in the study are averaged over the entire MidCaspian basin, while the observations are for the deep-water area.

3. Regarding the abstract, if possible, I think it is useful to have a more extensive abstract that includes some of the results mentioned in the conclusion. For example, the effect of regime shift of global climate on the different regions of the Caspian Sea.

4. Technical error, misspelling: Line 15, page 7: "compared to" instead of "comparted to"
* * *

---

## Referee Comment (RC2) · Anonymous Referee #2 · 6 Dec 2018

Review of the manuscript "Long-Term Evolution of the Caspian Sea Thermohaline Properties Reconstructed in an Eddy-Resolving OGCM" by Gleb S. Dyakonov and Rashit A. Ibrayev.

The manuscript analyses the decadal variability of the Caspian Sea thermohaline properties. A high-resolution ocean general circulation model is used including sea ice thermodynamics and air-sea interaction, forced by prescribed realistic atmospheric conditions and riverine runoff. The important outcome of the study is to find the reason of the rapid sea level rise ($\sim$2.5 m) between 1978-1995 that is depending on the variability of riverine and atmospheric forcing, rather than other any factors. Thus, authors results

are consistent with the commonly recognized theory, relating the Caspian Sea level fluctuations with global climate changes.

The abstract is compact containing the purpose of the study, the short methodology information, the most important results and the evaluation of the reconstruction experiments.

The scientific approach and applied methods are valid. It is well organized and accomplished by briefly reviewing some of the relevant literature and explaining how the current study is related to them beginning from earlier studies upto recent researches. Nevertheless, it is hard to follow the manuscript because of absent a map that showing name of the all geographic features (i.e. rivers, Karabogazgol etc.) that they are mentioned in the text. The weakness of the manuscript is that authors did not make any comparison between the model results and the temperature and salinity observations obtained in the Caspian Sea as they did for the sea level data. It is necessary to see the model results how agree with the observations in the point of view of model validation. On the other hand, it would be nice to put some circulation patterns at least one winter and one summer circulation together with temperature and salinity fields in the manuscript to show how good the model is by reconstruction the thermohaline properties. To add the circulation pattern before and after the period of climate shift would be also appropriate for the manuscript content.

Here is the list of suggested some corrections and changes:

- Adding the explanation of the coupling sigma and z coordinate systems in methodology section. - Line 29; instead "work (Dyakonov and Ibrayev, 2018)" is better to write "work of Dyakonov and Ibrayev (2018)" - Line 43; instead "In (Ibrayev, 2001; Ibrayev et al., 2001; Ibrayev et al., 2010)" is better to write" In Ibrayev (2001); Ibrayev et al. (2001) and Ibrayev et al. (2010)" - Line 57; instead "in (Ibrayev and Dyakonov, 2016; Dyakonov and Ibrayev, 2016) is better to change "in Ibrayev and Dyakonov (2016) and Dyakonov and Ibrayev (2016) - Line 59; same "in (Dyakonov and Ibrayev, 2018)"—> "in

Dyakonov and Ibrayev (2018)" - Line 81; same "in (Ibrayev et al., 2010)" —> "in Ibrayev et al. (2010)" - Line 85; same "in (Dyakonov and Ibrayev, 2018)" —> "in Dyakonov and Ibrayev (2018)" - Line 89; same "in (Schrum C. and Backhaus, 1999)" —> "in Schrum C. and Backhaus (1999)" - Line 286; same "in (Tuzhilkin et al., 2011)" —> "in Tuzhilkin et al. (2011)" - By Author contributions part, "The research was carried out by GD under the supervision of RI" GD and RI abbreviations are not known.

Please also note the supplement to this comment:
https://www.ocean-sci-discuss.net/os-2018-128/os-2018-128-RC2-supplement.pdf

---

## Referee Comment (RC3) · 11 Dec 2018

This paper studies the crucial issue of predicting climate-driven sea level changes in the Caspian Sea based on an appropriately designed numerical circulation model and interactive atmosphere-ocean fluxes. The specific question that is asked is if the seasonal - decadal sea-level and its large drift observed in response to climatic shift from late 1970's to the about 1990 can be investigated with a special hydrodynamic model subject to realistic forcing. The paper relies on earlier investigations of model sensitivity to certain physical processes that are of key importance in obtaining better predictions of climate-driven changes in sea-level.

The improvements on air-sea fluxes, bottom friction in shallow areas, initialization and spin-up and interaction of shallow waters with the deep sea are some of the issues that the authors have given care. It is shown that detailed eddy-resolving modeling with adequate fine-resolution representation of specific key processes and fluxes is able to produce and closely simulate the observed response of the Caspian Sea to both seasonal and climatic events. Yet information with sufficient detail is not given on how adjustments were made to tune the model with respect to identified key processes. For example, it would be desirable to know, from the reader's standpoint, what numerical values were selected and which parameterizations were used for bottom and internal friction, advection and diffusion schemes, and surface fluxes.

Some of the investigated long-standing questions are well resolved by this work, such as the relative roles of buoyancy and wind-driven circulations, inter-basin transports, shallow-deep sea interactions, winter-time convective mixing, as far as we know for the first time at such high resolution but not sufficiently emphasized by displaying these characteristics in some detail or in the conclusions. The paper is focused on climate response, but all the fine detail that finally achieves performance would be better appreciated if they could be better exposed and emphasized. For instance, surviving myth on total overturn of deep waters by severe winter convection seems finally to be settled by demonstration of limited penetration in the present period of investigation. However, remembering even greater excursions in past climates and consequent greater shifts in sea-level, it may be desirable to discuss in the last section of the paper if and how such more extreme changes could be expected or simulated by extension of the present results.

Similarly the roles of down-slope convection processes not represented in the model could be further discussed, from the points of view on short-term and climatic response, to elucidate issues in model development and prediction in the future. Other fine-scale processes such as fronts and upwelling could also be important in the climate sense although they are often considered to be short-term, as also shown earlier by

the authors, and they could be emphasized in their presentation and the discussion.

A minor note: The 6 year low-passed time-series plotted in red in Fig.2a-h is shifted by 6 years - which is the window length. If the low-pass should be centered there would be only a loss of 3 year at the beginning and end of the filtered series (and even this could be partially recovered by adjusting length near the ends). The accordingly corrected low-passed series should be presented in this Figure.

In order to help the authors with style and written language, editing changes are proposed on the pdf, which the authors could choose to adopt.

In short, the paper addresses an important problem of the Caspian Sea which is an important element of regional climate, and thus should be published with minor revisions placing emphasis on discussion of the results.

Please also note the supplement to this comment:
https://www.ocean-sci-discuss.net/os-2018-128/os-2018-128-RC3-supplement.zip

---

## Author Comment (AC1) · 2 Mar 2019

We thank the referee for his/her constructive comments and suggestions. Further the indicated remarks are discussed one-by-one. Attached is the revised manuscript provided in 2 versions for your convenience: with and without mark-up of changes (otherwise, the two documents are identical). Note: only substantial changes are marked-up in the attached manuscript.

1. Referee Comment: It is mentioned that the downsloping cascading process of cold saline waters along the slope of northern and eastern shelfs are not fully taken into account by the model. Is this due to using z-level grids in the model? If not, it would be

interesting to know how this process can be better resolved in numerical models.

Author Response: Yes, this process is rather poorly resolved in z-coordinate grid models. There are methods to better account for downsloping cascading, such as more sophisticated vertical coordinates and various parameterizations. However our attempts to apply some of those parameterizations in the Caspian Sea model had little effect, and this problem remains.

Changes in manuscript: Necessary clarifications were added in subsection "6.2 Middle Caspian".

2. Referee Comment: The model is validated against sea surface height at Baku and is shown to be able to well reproduce its long-term variability. However, for the salinity and temperature evolution, the comparison against observed data has not been shown (Although it is mentioned that the model is in good agreement with observations). It would be interesting to have a more detailed comparison of the model thermohaline structure against observations shown in Tuzhilkin and Kosarev, 2004. My understanding is that the model shows more salinity stratification in the period of 1960-1978 in the middle Caspian Sea. However, it is hard to reach a conclusion because the results shown in the study are averaged over the entire MidCaspian basin, while the observations are for the deep-water area.

Author Response: Indeed, the model shows somewhat greater salinity stratification in the Middle Caspian as a result of excessive surface freshening. This is one of the most significant model errors. We agree that the paper lacks T and S observational data for comparison and have added section "5 Model validation" including plots for Middle and Southern basins comparing T and S at 100 m in two locations with measurements data from Tuzhilkin and Kosarev (2004). More comprehensive validation against observational data would greatly expand the paper, so, as a compromise, only these two plots and sea level comparison are presented in this section. The paper is also supplemented with section "4 Surface circulation" to better explain the results

presented.

Changes in manuscript: Sections "4 Surface circulation" and "5 Model validation" were added.

3. Referee Comment: Regarding the abstract, if possible, I think it is useful to have a more extensive abstract that includes some of the results mentioned in the conclusion. For example, the effect of regime shift of global climate on the different regions of the Caspian Sea.

Author Response: Yes, the initial abstract appears to be overly short.

Changes in manuscript: The abstract was extended with main results presented in the paper.

4. Referee Comment: Technical error, misspelling: Line 15, page 7: "compared to" instead of "comparted to"

Author Response: Corrected, thank you.

Changes in manuscript: Misspelling corrected.

Please also note the supplement to this comment:
https://www.ocean-sci-discuss.net/os-2018-128/os-2018-128-AC1-supplement.zip
* * *

---

## Author Comment (AC2) · 2 Mar 2019

We thank the referee for his/her constructive comments and suggestions. Further the indicated remarks are discussed one-by-one. Attached is the revised manuscript provided in 2 versions for your convenience: with and without mark-up of changes (otherwise, the two documents are identical). Note: only substantial changes are marked-up in the attached manuscript.

1. Referee Comment: It is hard to follow the manuscript because of absent a map that showing name of the all geographic features (i.e. rivers, Karabogazgol etc.) that they are mentioned in the text.

[Figure]

Author Response: We agree that the paper lacks such map and have added geographical information on the fig. 1.

Changes in manuscript: Figure 1 was supplied with additional information.

2. Referee Comment: The weakness of the manuscript is that authors did not make any comparison between the model results and the temperature and salinity observations obtained in the Caspian Sea as they did for the sea level data. It is necessary to see the model results how agree with the observations in the point of view of model validation.

Author Response: Indeed, the paper lacks T and S observational data for comparison, so we have added section "5 Model validation" including plots for Middle and Southern basins comparing T and S at 100 m in two locations with measurements data from Tuzhilkin and Kosarev (2004). More comprehensive validation against observational data would greatly expand the paper, so, as a compromise, only these two plots and sea level comparison are presented in this section.

Changes in manuscript: Section "5 Model validation" was added.

3. Referee Comment: It would be nice to put some circulation patterns at least one winter and one summer circulation together with temperature and salinity fields in the manuscript to show how good the model is by reconstruction the thermohaline properties. To add the circulation pattern before and after the period of climate shift would be also appropriate for the manuscript content.

Author Response: We agree that visualization of the model solution would greatly help reader to follow the text and is necessary to explain some of the results presented. The paper is supplemented with section "4 Surface circulation" including 2D-plots of instantaneous sea surface salinity and temperature as well as monthly mean surface currents. The latter are presented for winter and summer and were averaged over two periods: before and after the climate regime shift, as suggested by the referee.

Changes in manuscript: Section "4 Surface circulation" was added.

[Figure]

4. Referee Comment: Referee suggested "Adding the explanation of the coupling sigma and z coordinate systems in methodology section"

Author Response: Coupling sigma and z coordinate systems is based on continuity of model solution and its z-derivative at the interface of the two systems. Thus it is rather straightforward and can be found by an interested reader in Ibrayev and Dyakonov (2016), referenced in the paper. In the present study we would like to refrain from describing such details of model design and focus on model results.

Changes in manuscript: None.

5. Referee Comment: Referee suggested changing the form of references, e.g.: "in (Tuzhilkin et al., 2011)" —> "in Tuzhilkin et al. (2011)", etc.

Author Response: We agree, the suggested form is preferable.

Changes in manuscript: The form of the references, mentioned by the referee, was changed accordingly.

6. Referee Comment: GD and RI abbreviations are not known

Author Response: GD and RI are the authors' initials (Gleb Dyakonov and Rashit Ibrayev). This form of the "Author contributions" section is standard for the Ocean Science Journal, though rather uncommon elsewhere.

Changes in manuscript: None.

Please also note the supplement to this comment:
https://www.ocean-sci-discuss.net/os-2018-128/os-2018-128-AC2-supplement.zip

---

## Author Comment (AC3) · 2 Mar 2019

We thank Dr. Özsoy for his constructive comments and suggestions. Further the indicated remarks are discussed one-by-one. Attached is the revised manuscript provided in 2 versions for your convenience: with and without markup of changes (otherwise, the two documents are identical). Note: only substantial changes are marked-up in the attached manuscript.

1. Referee Comment: The improvements on air-sea fluxes, bottom friction in shallow areas, initialization and spin-up and interaction of shallow waters with the deep sea are some of the issues that the authors have given care. It is shown that detailed

eddy-resolving modeling with adequate fine-resolution representation of specific key processes and fluxes is able to produce and closely simulate the observed response of the Caspian Sea to both seasonal and climatic events. Yet information with sufficient detail is not given on how adjustments were made to tune the model with respect to identified key processes. For example, it would be desirable to know, from the reader's standpoint, what numerical values were selected and which parameterizations were used for bottom and internal friction, advection and diffusion schemes, and surface fluxes.

Author Response: Comprehensive description of the model design would require considerable extension of the paper, so we discuss it only briefly. As for the bottom friction parameterization, it is based on the classical scheme from Weatherly & Martin, 1978. Its implementation is not straightforward, but its description would require describing also the entire model framework, and therefore it is omitted. Nonetheless, we agree that certain details of model design could be interesting for a reader and should be added.

Changes in manuscript: Section "2.1 Model description" was supplemented with details of the parameterizations and schemes used in the model, including particular numerical values. In section "2.2 External forcing" the performed corrections of atmospheric forcing were specified. Correction of solar radiation is no longer mentioned in this section, as in this particular experiment this flux was not altered (which was overlooked when preparing the initial draft).

2. Referee Comment: Some of the investigated long-standing questions are well resolved by this work, such as the relative roles of buoyancy and wind-driven circulations, inter-basin transports, shallow-deep sea interactions, winter-time convective mixing, as far as we know for the first time at such high resolution but not sufficiently emphasized by displaying these characteristics in some detail or in the conclusions. The paper is focused on climate response, but all the fine detail that finally achieves performance would be better appreciated if they could be better exposed and emphasized. For

instance, surviving myth on total overturn of deep waters by severe winter convection seems finally to be settled by demonstration of limited penetration in the present period of investigation. However, remembering even greater excursions in past climates and consequent greater shifts in sea-level, it may be desirable to discuss in the last section of the paper if and how such more extreme changes could be expected or simulated by extension of the present results.

Author Response: The circulation patterns of the Caspian Sea are extremely diverse and all of the main features cannot be considered in detail within one paper. In this particular study we aim to investigate the very possibility of modeling the long-term variability of the sea thermohaline circulation as well as the role of various factors of its formation. Nonetheless, we agree that some of the results discussed in the paper could be better explained by visualizing the circulation obtained in the model. Thus, we have added section "4 Surface circulation" containing 2D-plots of surface currents, temperature and salinity, useful for understanding the behavior of space-averaged parameters, discussed in further sections. Section "5 Model validation" was also added, which provides additional analysis along with verification of the model results. As for possible more extreme changes, in our opinion the paper does not present sufficient basis for such forecasts, but could be considered as another step towards profound understanding of climate change impact on lakes and isolated seas.

Changes in manuscript: Sections "4 Surface circulation" and "5 Model validation" were added.

3. Referee Comment: Similarly the roles of down-slope convection processes not represented in the model could be further discussed, from the points of view on short-term and climatic response, to elucidate issues in model development and prediction in the future. Other fine-scale processes such as fronts and upwelling could also be important in the climate sense although they are often considered to be short-term, as also shown earlier by the authors, and they could be emphasized in their presentation and the discussion.

Author Response: The role of down-slope convection was not investigated in the paper: its neglecting was only suggested as a possible reason of accumulating model error in the abyssal waters of the sea. To clarify why this process is not well represented in the model a short explanation was added. Fine-scale processes such as fronts and upwelling play, indeed, an important role in the Caspian thermohaline regime, including in the long-term. To cover this subject a bit more some of the main circulation patterns are additionally presented in section "4 Surface circulation". However, in-depth discussion of their influence is also beyond the scope of the paper.

Changes in manuscript: A short elucidation of the model design with respect to down-slope convection was added in section "6.2 Middle Caspian". Section "4 Surface circulation" was added to better consider some of the fine-scale processes: upwelling, frontal intrusions, jet currents.

4. Referee Comment: A minor note: The 6 year low-passed time-series plotted in red in Fig.2a-h is shifted by 6 years - which is the window length. If the low-pass should be centered there would be only a loss of 3 year at the beginning and end of the filtered series (and even this could be partially recovered by adjusting length near the ends). The accordingly corrected low-passed series should be presented in this Figure.

Author Response: Indeed, this is a good idea. The 6 year low-passed time-series plotted in red in fig.2 was replaced by a 5-year centered moving average, which eliminated phase shift.

Changes in manuscript: Figure 2 was changed.

5. Referee Comment: In order to help the authors with style and written language, editing changes are proposed on the pdf, which the authors could choose to adopt.

Author Response: The proposed textual changes were adopted. Thank you very much!

Changes in manuscript: Numerous textual changes, suggested by the referee, were made, without any substantial changes with respect to the paper contents.
Please also note the supplement to this comment:
https://www.ocean-sci-discuss.net/os-2018-128/os-2018-128-AC3-supplement.zip
* * *

---

## Author Response (AR2)

**Long-Term Evolution of the Caspian Sea Thermohaline Properties Reconstructed in an Eddy-Resolving OGCM**

Gleb S. Dyakonov1,2, Rashit A. Ibrayev1,2,3

1Northern Water Problems Institute, Russian Academy of Sciences, Petrozavodsk, Russia 2Shirshov Institute of Oceanology, Russian Academy of Sciences, Moscow, Russia

3Marchuk Institute of Numerical Mathematics, Russian Academy of Sciences, Moscow, Russia

Correspondence to: Gleb S. Dyakonov (gleb.gosm@gmail.com)

Abstract. The dDecadal variability of the Caspian Sea thermohaline properties is investigated by meansusing of a highresolution ocean general circulation model including sea ice thermodynamics and air-sea interaction, forced by prescribed realistic atmospheric conditions and riverine runoff. The model describes synoptic, seasonal and climatic variations of the sea thermohaline structure, water balance and a sea level levelheight. A reconstruction experiment was conducted for the period of 1961-2001, - covering a major regime shift in the global climate of during 1976-1978, which allowed investigating the Caspian Sea response to such significant episodes of climate variability. The model reproduced sea level evolution reasonably well despite that many factors (such as possible seabed changes and yet insufficiently explored underground water infiltration) were not taken into account in the numerical reconstruction, which allows to investigate the Caspian Sea response to such significant episodes of climate change... This supports the hypothesis relating rapid Caspian Sea level rise in 1978-1995 with the global climate change, which caused variation of local atmospheric conditions and riverine discharge reflected in the used external forcing data, as is shown in the paper. Other effects of the climatic shift are investigated including a decrease of salinity in the active layer, strengthening of its stratification and corresponding diminishing of convection. It is also demonstrated that water exchange between the three Caspian basins (northern, middle and southern) plays a crucial role in the formation of their thermohaline regime. The reconstructed long-term trends in the sea water salinity (general downtrend after 1978), temperature (overall increase) and density (general downtrend) circulation patterns are studied, including considered with an assessment of the the influence of main surface circulation patterns and model error accumulation.

25

5

10

15

20

**1** Introduction**

The Caspian Sea is the largest enclosed water body on earth, covering with a surface area of more than over 370 000 km2, and has a catchment area, that is almost 10 times greater. Yet it is highly sensitive to variations in the global and elimate system as well as regional climate systems as well as the regulation of river runoff and other economic activities that include major schemes of river regulation in the region. This is vividly reflected in the evolution of the Caspian Sea level, which is

subject to large fluctuations both on seasonal and decadal timescales. The water balance of the isolated sea varies significantly due to the seasonal character of the riverine discharge, accounting for sea which accounts for level oscillations with an an amplitude amplitude of 20–40 cm. Long-term fluctuations of the level are even larger: in the second half of the 20th century they amounted to 2.5 m.

Prediction of the long term impacts of global climate changes and man-made activities on the Caspian in the long term 35 represents a great scientific challenge and is an important task for fisheries, coastal development and other industries of the region. Ocean general circulation models (OGCM) have greatly advanced our understanding of the Caspian Sea circulation patterns, particularly its seasonal variability (Arpe et al., 1999; Ibrayev, 2008; Kara et al., 2010; Ibrayev et al., 2010, GunduzGündüz and Özsov, 2014, Diansky et al., 2016). The increasingFurthermore, production of global atmospheric reanalysis datasets and their availability over several decades have, covering extended periods of time (usually decades). 40 made possible a retrospective studies study of the long-term evolution of athe marine environment, based on numerical reconstruction of its response to external forcing, as will be done in which is a subject of the present paper. This approach was applied in our previous work (Dyakonov and Ibrayev, 2018) with the emphasis on the long-term variability of the Caspian Sea water balance and its sensitivity to external factors. Now we use the same model to study the evolution of thermohaline properties (temperature, salinity and density) of the Caspian Sea during in 19601-20001. The period is 45 particularly interesting, as it covers one of the most notable events of global climate change – the climate shift of 1976-1978, also referred to as the Great Pacific Climate Shift, widely discussed in literature (Miller et al., 1994; Wooster and Zhang, 2004; Powell and Xu, 2011). The shift was associated with a change in major<del>of many</del> climatic indicators such as<del>processes</del>. including the North Atlantic Oscillation with, which led to a significantly increased 
[revised manuscript text omitted]